# Health Expenditure, Institutional Quality, and Under-Five Mortality in Sub-Saharan African Countries

**DOI:** 10.3390/ijerph21030333

**Published:** 2024-03-12

**Authors:** Kin Sibanda, Alungile Qoko, Dorcas Gonese

**Affiliations:** Department of Business Management and Economics, Faculty of Economics and Financial Sciences, Walter Sisulu University, Mthatha Campus, Mthatha 5700, South Africa; kinsibanda@wsu.ac.za (K.S.); aqoko@wsu.ac.za (A.Q.)

**Keywords:** health expenditure, institutional quality, under-five mortality rate, GMM, sub-Saharan African countries

## Abstract

The aim of this study is to examine the relationship between health expenditure, institutional quality, and under-five mortality rates in sub-Saharan African countries. Specifically, the study seeks to explore the mediating role of institutional quality in this relationship, focusing on understanding how variations in healthcare spending and institutional frameworks impact child health outcomes. By examining these dynamics, the study aims to provide valuable insights that can inform evidence-based policy interventions to reduce under-five mortality and improve child health outcomes in the region. Utilizing data spanning the years 2000 to 2021 from 46 sub-Saharan African countries, this study employs a systems GMM model to explore the intricate relationship between health expenditure and under-five mortality rates (U5MRs), with a particular focus on the mediating role of institutional quality. The findings reveal that the quality of institutions significantly influences the impact of health expenditures on the U5MR. Strong institutional quality enhances the effectiveness of health expenditure in improving child health outcomes, particularly concerning the allocation of external health funds. Conversely, poor institutional quality amplifies the positive impact of domestic private and out-of-pocket health expenditures on the U5MR, as these serve as coping mechanisms in the absence of robust public healthcare systems. This research emphasizes the need for strategies that increase health expenditure and prioritize institutional strengthening to ensure efficient resource allocation and healthcare system management, thereby reducing under-five mortality rates. Furthermore, it underscores the importance of policies that minimize reliance on private and out-of-pocket health expenditures, which can lead to financial burdens and worsened health outcomes. Sub-Saharan African countries can make significant strides toward improving child survival and overall public health by addressing these issues.

## 1. Introduction and Background

In spite of the remarkable improvements in health conditions and status worldwide, sub-Saharan countries still suffer from the worst health challenges [1,2,3,4,5]. Under-five mortality has emerged as a critical challenge in the sub-Saharan region [6,7]. According to the UN IGME (2018) and Alimi and Ajide (2021) [8,9], the under-five mortality rate in the sub-Saharan region has been higher compared to many other regions worldwide. For instance, SSA has recorded an average of 60 deaths per 1000 births, while other regions such as MENA, Southeast Asia, Latin America, South America, North America, and Europe recorded averages of 18, 22, 18, 16, 17 and 14 deaths per 1000 births, respectively, in 2021.

Health spending in sub-Saharan Africa has shown improvement between 2000 and 2021, but the extent of improvement varies from country to country within the region. Sub-Saharan African countries such as Kenya, Ethiopia, Uganda, Ghana, Rwanda, South Africa, Malawi, and Tanzania have received substantial international aid and donor support for health-related initiatives [10,11]. The sub-Saharan countries, particularly South Africa, Tanzania, Botswana, Rwanda, Ghana, Kenya, Senegal, and Malawi, have also improved investments in healthcare infrastructure, training of healthcare personnel, and the expansion of health services, which also played a role in increasing health spending [12,13,14].

In sub-Saharan Africa, where healthcare resources are often scarce and disparities are pronounced [15,16], the relationship between health expenditure and under-five mortality rates has emerged as a critical concern, reflecting the complex interplay of various factors. By focusing on under-five mortality rates, we aim to comprehensively understand the broader spectrum of child mortality within the sub-Saharan African context. Under-five mortality rates serve as a widely recognized indicator of child health and well-being, reflecting not only infant and neonatal mortality but also mortality rates for children up to five years of age [16]. One of the pivotal determinants in this relationship is the quality of institutions. The institutional quality in sub-Saharan Africa between 2000 and 2021 has been characterized by positive developments and ongoing challenges. Some countries (Botswana, Mauritius, Ghana, Cape Verde, Senegal, Rwanda, Kenya, Zambia, and South Africa) in sub-Saharan Africa made progress in improving governance control of corruption and political stability [17]. However, in countries like Nigeria, DRC, Zimbabwe, Uganda, Angola, and Cameroon, inconsistencies in applying the rule of law, corruption within legal institutions, and slow justice systems remained challenging [17,18].

Even if health expenditure and institutional quality have improved in other sub-Saharan African countries, the region remains the home to a high infant mortality rate [7]. Accordingly, the under-five mortality rate serves as a stark measure of child well-being, reflecting the availability and effectiveness of healthcare interventions, the prevalence of preventable diseases, and the socio-economic status of communities. Thus, the primary concern is how much health spending has influenced the outcome, particularly under-five mortality in the region. Empirical literature [19,20,21,22] has documented evidence of the relationship between health expenditure and child mortality. Yet, there is still no consensus on how health expenditure influences child mortality, particularly the under-five mortality rate. While numerous studies [5,23,24,25,26,27,28] have been conducted on health expenditure and under-five mortality in sub-Saharan Africa, there are several justifications for the need to consider the mediating effects of institutional quality and incorporate all relevant health expenditure indicators.

This paper examines the relationship between health expenditure and under-five mortality, spotlighting the pivotal role that institutional quality plays in shaping outcomes across the diverse countries of sub-Saharan Africa. The paper provides valuable insights for policymakers by highlighting the relationship between different indicators of health expenditure and under-five mortality. This informs decisions about where to allocate resources for healthcare, emphasizing the importance of public spending to ensure access to essential services. In addition, considering the role of institutional quality in this context can help identify the importance of good governance, transparent policies, and effective healthcare institutions. By examining these multifaceted relationships, the current paper uncovers the hidden synergies, trade-offs, and dependencies that are key to improving child health outcomes and healthcare resource allocation. The paper employs the Systems Generalized Method of Moments (SGMM) to address endogeneity and cross-sectional dependence commonly observed in panel data. The present study contributes to the broader goal of achieving sustainable development and improved child health outcomes in sub-Saharan countries.

The following section presents the related literature on health expenditure and under-five mortality. The third section shows the methodology used to assess the said relationship. The fourth section provides the empirical findings of the current study. The last section concludes the paper.

## 2. Literature Review on Under-Five Mortality

### 2.1. Theoretical Literature

The relationship between health expenditure and under-five mortality can be explained by the human capital [29,30] and the health production function theories. The human capital theory postulates that individuals make decisions about their health and healthcare utilization based on income, education, institutions, and health behaviors [31,32]. The health production function, on the other hand, treats health as an output produced from various inputs, including healthcare services, health expenditure, lifestyle choices, and socioeconomic factors. This paper considers institutional quality as the mediating factor in the under-five mortality-health expenditure nexus. The said relationship is explained by the institutional theory, which postulates that formal and informal rules, norms, and practices shape the behavior of healthcare organizations. The theory advocates that institutional quality, such as the rule of law, regulatory quality government effectiveness, political stability, and control of corruption, can affect health expenditure by determining the availability of healthcare services and the utilization of resources, which ultimately influence health outcomes. The human capital and health production function theories consider health expenditure and institutional quality as inputs into healthcare. Thus, the current paper follows the Grossman (1972) [29] approach in examining the impact of health expenditure and institutional quality on under-five mortality in SSA countries.

### 2.2. Empirical Literature

Health expenditure has long been viewed as a potential complement to improving health outcomes such as child survival. Yet there is still controversy over how effective the health expenditure is on the under-five mortality rate (U5MR). Different conclusions have been drawn, as some studies document evidence of no association between health spending and the U5MR [27,33] while other studies indicate a positive [5,34] or negative [35,36,37,38] relationship. Public health expenditure has been a commonly used indicator in most of the literature on health spending and health outcomes. Thus, studies that consider different health expenditure indicators found disparate effects of health expenditure on the U5MR. For example, Logarajan et al. (2022) [39] considered three indicators, including public, private, and out-of-pocket health expenditures. Their ARDL estimation technique indicates that out-of-pocket health expenditure reduces the U5MR, while public and private health expenditures are insignificant to explain changes in the U5MR in Malaysia.

Numerous studies that assess the direct effect of health spending, particularly public health expenditure, support health expenditure as an U5MR-reducing factor. Using a set of 158 developing and developed countries, Hadipour, Delavari, and Bayati (2023) [40] found a negative effect of health expenditure on the U5MR. Dhrifi (2020) [41] found similar results when using the generalized methods of moments (GMM) to assess the impact of health expenditure in 93 developed and developing economies. More so, the fixed effect in Owusu, Sarkodie, and Pedersen (2021) [38] indicates that healthcare expenditure reduced the U5MR in middle and lower-income countries between 2007 and 2008. Consistently, Ahmad and Hasan (2016) [35], Rahman and Khanam (2018) [36], and Kato et al. (2018) [6] document evidence of a negative and significant impact of health expenditure on the U5MRs in Malaysia, Uganda, and ASEAN countries, respectively. There are also contrasting findings; for instance, Kulkarni (2016) [42], Azuh et al. (2020) [34], and Logarajan et al. (2022) [39] found a significant and positive effect of health expenditure on under-five mortality in BRICS countries, Nigeria and Malaysia, respectively. Studies that assessed the impact of health expenditure on under-five mortality in SSA focused on public health expenditure as the common indicator of health spending. Only a few studies considered other indicators, such as private [24,43], out-of-pocket [44] and external health expenditure [27]. Ashiabi, Nketiah-Amponsah, and Senadza (2016) [24] and Arthur and Oaikhenan (2017) [43] assessed the impact of public and private health expenditures on health outcomes in 40 SSA countries using a fixed effect estimation technique. The studies are in consensus that public health expenditure reduces five mortality rates and that private health expenditure is insignificant in explaining changes in under-five mortality rates. Similar findings were found by Kiross et al. (2021) [27], who argued that public and external health expenditures improve health outcomes while private health expenditure has an insignificant impact. Moreover, Makuta and O’Hare (2015) [23], Chewe and Hangoma (2020) [25] and Ayipe and Tanko (2023) [28] supported the direct negative effect of public health expenditure in SSA. The conflicting findings are found in the literature on health expenditure on the U5MR in SSA. Using panel data from 200 to 2008 in ten SSA countries, Akinlo and Sulola (2019) [5] found that health expenditure worsens under-five mortality.

Although Ouedraogo, Dianda, and Adeyele (2020) [45] support that institutional quality is paramount in healthcare efficiency, most studies [28,38,46] focus on the direct impact of health expenditure on U5MRs. Nevertheless, studies that considered institutional quality as the mediating effect in the said relationship only focused on public health expenditure. Thus, studies in Malaysia [35], WAEMU [47], developing and developed countries [41], high and low-income countries [40], and in SSA [23,26] are in consensus that the quality of governance enhances the public health expenditure to reduce the U5MR.

The above exposition demonstrates that studies [5,28,46] on health expenditure and under-five mortality focused mostly on public health. A few [24,27] consider the effect of private and public health expenditure on under-five mortality in SSA. The current study assesses the impact of health expenditure on under-five mortality in SSA countries, considering four indicators: public health, private health, out-of-pocket, and external health expenditure on under-five mortality. Most studies [5,28,43,46] focused on the direct effect of health expenditure on under-five mortality.

Again, some studies [45] assessed only the effect of institutional quality on five mortality, while a few studies [23,26] consider institutional quality as the mediating factor in the under-five mortality-health expenditure nexus. However, the current paper considers the institutional quality’s impact on the effect of each of the four health expenditure indicators on under-five mortality in SSA. In doing so, the present form tests the hypothesis that there is no significant direct and indirect impact of health expenditure (public, private, out-of-pocket, and external) on the under-five mortality rate in sub-Saharan Africa. The alternative hypothesis is that health expenditure (public, private, out-of-pocket, and external) has a significant direct and indirect impact on the under-five mortality rate in sub-Saharan Africa.

## 3. Materials and Methods

### 3.1. Data Sources

The data for the study were obtained from the Global Economy, World Bank: World Development Indicators websites. The paper used annual panel data for 46 sub-Saharan African (SSA) countries (see list in Appendix G) from 2000 to 2021.

### 3.2. Econometric Model

In our empirical estimations of the nexus between health expenditure, institutional quality, and under-five mortality for SSA countries, under-five mortality was used as the dependent variable. The study used a panel data estimation technique to examine the impact of health expenditure on the under-five mortality rates (U5MRs) in 46 sub-Saharan Africa countries from 2000 to 2021. The study adopted and modified the model by Ayipe and Tanko (2023) [28], who examined the relationship between public health expenditure and under-five mortality in low-income sub-Saharan African countries. The model was formally specified as Equation (1) below.
(1)$${u5mr}_{it}=\alpha_{i}+\beta_{1}{HED}_{it}+\beta_{2}{TFR}_{it}+\beta_{3}{PPF}_{it}+\beta_{4}{ROD}_{it}+\beta_{5}{DPT}_{it}+\;\beta_{6}{TPOD}_{it}+\varepsilon_{it}$$
where $\alpha_{it}$ is the specific factors of country *i*; $\beta_{s}$ represents the slope coefficients to be estimated; ${u5mr}_{it}$ is the under-five mortality rate of country *i* at time *t*; ${HED}_{it}$ represents health spending from domestic funds as a proportion of GDP for country *i* at time *t*; ${TFR}_{it}$ is the total fertility rate in country *i* at time *t*; ${PPF}_{it}\;$is the percentage of the population who are female in country i at the time t; ${ROD}_{it}$ is the percentage of rural people who are defecating in the open (as a percentage of rural people) in country *i* at time *t*; ${DPT}_{it}$ is the child immunization rate against DPT in country *i* at time *t*; ${TPOD}_{it}$ is the percentage of the total population of people defecating in the open (as a percentage of the total population) in country *i* at time *t*; and $\varepsilon_{it}$ is the stochastic disturbance term. The model for this study is presented below in Equation (2) as follows:(2)$${u5mr}_{it}=\alpha_{i}+\beta_{1}{HE}_{it}+\beta_{2}{INST}_{it}+\beta_{3}{HE*INST}_{it}+\beta_{4}X_{it}+\varepsilon_{it}$$
where $\alpha_{it}$ is the specific factors of country *i*; $\beta_{s}$ represents the slope coefficients to be estimated; ${u5mr}_{it}$ is the under-five mortality rate of country *i* at time *t*; and ${HE}_{it}$ represents health expenditure for country *i* at time *t*. The paper considered four indicators of health expenditure, which included public health expenditure (*phe*), domestic private health expenditure (*hedp*), out-of-pocket health expenditure (*heoup*), and external health expenditure (*heext*). ${INST}_{it}$ represents the institutional index computed in Stata software using principal component analysis (PCA) (see Appendix F) from six indicators of institutional quality, including the rule of law, government effectiveness, control of corruption, regulatory quality, voice accountability, and political stability data [47]. The values of each of the six components range from −2.5 weak to 2.5 strong [48]. Thus, the PCA effectively merges the six institutional indices (see Appendix D) into a single composite index as recommended by Karamizadeh et al. (2013) [49], Dutta, Gupta and Sengupta (2019) [50], Ouedraogo, Dianda and Adeyele (2020) [45], Kouton, Bétila and Lawin (2021) [51], and Hadipour, Delavari and Bayati (2023) [40]. The PCA reduces the number of variables in a dataset while retaining the most essential information [52]. It identifies the principal components, which are linear combinations of the original variables that capture the maximum amount of variance in the data information [49,53]. As a result, the PCA simplifies the analysis and can improve computational efficiency [54].

The paper also uses the PCA to construct a weighted ICT index (*ict*) from the three components of ICT, including fixed broadband subscription (*fbs*), internet usage (*itu*), and mobile phone use (*mcs*). The three ICT dimensions were also used in previous literature [50,51]. The full definitions of the individual components of the institutional quality (INST) and ICT indices are presented in Appendix D.

${INST*HE}_{it}$ denotes the interaction term, which is the multiplication of the institutional quality variable with each health expenditure indicator (*pheinst, hedpinst, heoupinst,* and *heextinst*).$\;X_{it}$ is the vector of all explanatory variables, which includes the prevalence of HVI (*phiv*), prevalence of undernourishment (*pun*), child immunization (*imm*), maternal mortality (*mam*), secondary education (*senr*), GDP per capita (*gdppc*), and $\varepsilon_{it}$ is the stochastic disturbance term.

Although the fixed effects estimation is likely to control for unobserved fixed effects, the endogeneity problem persists because of dynamism and unobserved time-varying omitted variables. The under-five mortality model usually consists of explanatory variables (such as maternal health) that affect the *U*5MR and can also be affected by the U5MR, leading to reverse causality and endogeneity biases (Hu and Mendoza (2013) [55]. To account for these issues, the paper employs dynamic panel data estimation techniques that control for endogeneity bias [56]. Dynamic panel data estimators, such as generalized methods of moments (GMM) [57] and panel autoregressive distributed lag (PARDL) [58], are commonly used estimation techniques to control endogeneity bias. However, the usage of these estimation methods is determined by the characteristics and status of the data [59].

### 3.3. Econometric Techniques

Based on the characteristics of the data (N > T) used in the study, the current paper considers the dynamic panel GMM estimation technique. This follows Roodman (2009) [59] and Asteriou and Hall (2016) [56], who argued that GMM is particularly suitable and efficient when the number of cross-sections (N = 46) is greater than the number of periods (T = 22). The GMM estimation technique is a generic method for estimating parameters in statistical models that use instruments that are functions of the model parameters and the data such that their expectations are zero at the parameter’s true value [57,59]. The estimation technique controls for the endogeneity of the lagged dependent variable in a dynamic panel model when there is a correlation between the explanatory variable and error term in a model [56,57]. Thus, the GMM uses the exogenous instrumental variables (IV) to surmount the endogeneity. This implies that using the GMM estimator makes the model more flexible, consistent, and efficient, given a richer set of instruments [60]. The dynamic panel GMM equation for the U5MR-health expenditure model can be written in the following specification:(3)$${lU5MR}_{it}={\Phi lU5MR}_{it-1}+{\beta_{1}HE}_{it}+{\beta_{2}HE}_{it}*{INST}_{it}+{{\vartheta X}_{it}+\gamma Z^{\prime }}_{it}+\delta_{i}+\omega_{t}+\varepsilon_{it}$$
where ${lU5MR}_{it}$ is the log of the under-five mortality rate, and it is the dependent variable, ${lU5MR}_{it-1}$ is the lagged dependent variable, Φ is the parameter of the lagged dependent variable, HE denotes health expenditure, which is the main explanatory variable, and $X_{it}$ represents other explanatory variables.

$Z_{it}^{\prime }$ is the vector for control variables and is assumed to comprise a set of variables highly correlated with the explanatory variable but orthogonal to the error term, δ is the unobserved country-specific fixed effects, ω represents time effects, β and γ are parameters, i is the number of cross-sections (=1…….,N), t is the number of time series (=1…….,T), and ε is the disturbance term. ${INST*HE}_{it}$ denotes the interaction effect of health expenditure at the specified level of institutional quality, which is derived from Equation (3) as:(4)$$\frac{\partial LHE}{\partial LU5MR}=\beta_{1}+\beta_{2}*{INST}_{it}$$
where $\beta_{2}$ denotes the extent to which institutional quality moderates the effect of health expenditure (including public, private, out-of-pocket, and external health expenditure) on the U5MR in SSA.

The GMM consists of differenced GMM (DGMM) [61] and system GMM (SGMM) [57,62]. The differenced GMM (DGMM) corrects endogeneity by transforming all the explanatory variables by differencing and removing fixed effects [61]. However, the DGMM’s weakness is subtracting previous observations from contemporaneous ones, thereby magnifying gaps in unbalanced panels [57]. Thus, applying the DGMM in unbalanced panel data will likely produce weak, inconsistent, and inefficient estimation results [59]. According to Blundell and Bond (1998) [57], the DGMM could produce biased and inadequate estimates of the parameter of the estimated lagged dependent variable (Φ) if poor instruments are applied. Yet the SGMM subtracts the average of a variable’s future available observations, thereby minimizing data loss.

The SGMMs are more desirable because they correct endogeneity by introducing more instruments to improve efficiency [61], meaning they are efficient and consistent when there is a correlation between the explanatory variable and the error term in the regression model. Also, Roodman (2009) [63] suggests that the SGMM uses orthogonal deviations instead of subtracting the previous observations from the contemporaneous ones, thus subtracting the average of all future available observations of a variable. Therefore, the most preferred GMM estimator is the SGMM, which is more desirable than the differenced GMM (DGMM). Therefore, the alternative estimator to use is the SGMM estimator. The study also employs the Bond, Hoeffler, and Temple (2001) [64] rule of thumb to determine the appropriate method between DGMM and SGMM. According to Bond et al. (2001) [64], if, after using the DGMM, the coefficient is lower than or very close to the fixed effect (DGM M≥ FE) coefficient, then the system GMM (SGMM) will be appropriate and efficient to use and vice versa. This means that if the DGMM estimate obtained is close to or lower than the FE, the DGMM is downward biased mainly because of weak instrumentation [63], so the SGMM should be the preferred estimation technique. The following section presents the definition and source of the variables of concern.

### 3.4. Variable Definitions, Expected A Priori, and Sources

The measurement of variables and data sources are given in Table 1 below.

The paper employs the four indicators of health expenditure based on Farag et al. (2013) [65], Kiross et al. (2021) [27], and Owusu, Sarkodie, and Pedersen (2021) [38] who indicated that the indicators of health spending have a significant effect on child mortality. Farag et al. (2013) [65] and Kiross et al. (2021) [27] documented evidence of the impact of public health expenditure on child mortality, and a positive but insignificant impact of private and external health expenditure. Thus, the study expects both positive and negative effects of health expenditure indicators on under-five mortality rates in SSA countries. Various studies [27,41,51,65,66,67,68] document an adverse effect of immunization on secondary education, ICT, GDP per capita, and institutional quality on under-five mortality rates. Therefore, the said variables are expected to have a reducing effect on under-five mortality rates.

According to Chihana et al. (2013) [69], Mutabazi, Zarowsky, and Trottier (2017) [70], and Tlou, Sartorius, and Tanser (2018) [71] the prevalence of HIV in mothers is a significant risk factor associated with under-five mortalities. Thus, a positive effect of the prevalence of HIV on the U5MR is expected. Studies such as Panagariya (2013) [72], Uribe-Quintero et al. (2022) [73], and Gamal et al. (2023) [74] suggest that undernourishment has an adverse effect on under-five health outcomes. As a result, the current study expects a positive relationship between the prevalence of undernourishment and under-five mortality rates. According to Finlay et al. (2015) [75] and Moucheraud et al. (2015) [76], maternal mortality reduces the chances of survival in children under five. Thus, maternal mortality is expected to positively affect under-five mortality in SSA countries.

### 3.5. Diagnostic Tests

As for the GMM, the appropriateness of the model and selection of the variable to be used in the model are determined by the diagnostic tests, which include the number of groups compared to the number of instruments, the AR (1) and AR (2), which test for first and second order correlation, and the Hansen and Sargan tests, which test for validity or overidentification of the instruments [57,63]. The p-vales of the AR (1), AR (2), and Hansen and Sargan tests are expected to be insignificant and confirm the absence of first and second-order correlation and the validity of the instruments, respectively [63].

## 4. Empirical Results

The descriptive statistics for the variables used in the study are presented in Table 2.

### 4.1. Descriptive Statistics

Table 2 shows that the standard deviations are large enough to explore variance in the data for the variables of concern. Table 2 indicates that the average U5MR value between 2000 and 2021 is 90.2 per 1000 births. Table 2 suggests that the public health expenditure (PHE) has a lower (5.2%) average value, while the domestic private health expenditure has the highest average value (48.02%) in SSA countries between 2000 and 2021. Again, descriptive statistics show that health spending is primarily out-of-pocket, with an average of 40.5% between 2000 and 2021. Furthermore, Table 2 demonstrates that 28.24% of health spending in SSA countries is external. The average HIV prevalence and undernourishment rates were 5% and 21.7%, respectively. According to Table 2, 75% of children in SSA nations received an immunization vaccine before reaching one year. Between 2000 and 2021, the average value for secondary education enrolment is 44.3%. The average ICT indices (FBS, ITU, and MCS) are 0.99%, 12.1%, and 49.4/100 people, respectively. The average GDP per capita is $2120. Between 2000 and 2021, the average point for the institutional quality index was −0.004.

**Table 2 ijerph-21-00333-t002:** Descriptive Statistics.

Variable	Obs	Mean	Std. Dev.	Min	Max
U5MR	1034	90.12186	42.54212	14	229
phe	956	5.224079	2.264309	1.26	20.41
hedp	956	48.01509	18.64855	8.34748	87.93836
heoup	956	40.44704	20.83885	2.993242	84.18211
heext	953	28.24149	32.24391	0	228.0008
phiv	990	5.00798	6.693593	0.1	29.8
pun	720	21.71611	13.81633	3.1	70.9
imm	1034	75.87911	18.74111	19	99
mam	987	502.4762	290.1703	3	1682
senr	620	44.2679	23.2978	5.51	114.71
fbs	743	0.9869448	3.404632	0	38.77
itu	1002	12.09936	16.17075	0.01	81.59
mcs	1026	49.37572	42.79125	0	185.56
ict	725	0.3427037	1.511278	−1.270262	10.10088
gdppc	998	2120.417	2785.312	255.1	16,747.34
inst	986	−0.0039528	2.212385	−6.015036	5.546552

Source: Author’s Computations from Stata Descriptive Statistics Regressions.

The correlation coefficient matrix (see Appendix A) shows that the under-five mortality rate is negatively correlated with public and external health expenditure while positively correlated with domestic private health and out-of-pocket health expenditure. The U5MR is also negatively correlated with secondary education enrolment, ICT, immunization, GDP per capita, and institutional quality. However, the prevalence of HIV undernourishment and maternal mortality are positively correlated with under-five mortality rates in SSA countries. Again, the variance inflation factor (VIF) values (see Appendix B) of the explanatory variable are less than 5 or 10, which implies the absence of multicollinearity in the analysis of the independent variable components. This also aligns with the recommendation of Gujarati and Porter (2010) [77] and Asteriou and Hall (2016) [56] The correlation coefficient matrix and the VIF test indicate that all the explanatory variables are not linearly dependent. They may thus be used in the same model of regression.

The Im Pesaran and Augmented Dickey–Fuller (ADF) stationarity test results (see Appendix C) indicate that variables such as out-of-pocket health expenditure, prevalence of HIV, prevalence of undernourishment, ICT, and GDP per capita are stationary after the first difference. At the same time, the rest of the variables are stationary at levels. Thus, the paper applied the differencing transformation for the variables that are stationary after first differencing, as recommended by Johnston and DiNardo (1997) [78], Brooks (2008) [79], Greene (2010) [80], and Asteriou and Hall (2016) [56].

Determination tests of the appropriate estimator between DGMM and SGMM were performed on each health expenditure indicator, based on the coefficients of the lagged dependent variables (Bond et al., 2001) [64]. Thus, each model’s coefficients of the lagged dependent variables are summarized in Appendix E. The results (see Appendix E) of the DGMM coefficients are lower than those of fixed effect (FE), indicating that the SGMM is the appropriate and efficient estimation technique for the study.

### 4.2. Empirical Results

This section discusses the empirical results of the study in Table 3 and Table 4. Table 3 presents empirical results of the direct effect of health expenditure on the U5MR, and Table 4 shows the impact of institutional quality on the U5MR-health expenditure nexus in SSA countries. In this study, the impact of health expenditure is divided into four categories, including public health expenditure (*PHE*), domestic private health expenditure (*HEDP*), out-of-pocket health expenditure (*HEOUP*), and external health expenditure (*HEEXT*). Model 1 in Table 3 is the benchmark model that shows empirical results of the direct impact of public health expenditure on the *U5MR* in SSA. Models 2 through 4 show the effects of *HEDP*, *HEOUP*, and *HEEXT* on the *U5MR*, respectively.

Table 3 indicates that the under-five mortality rate for the previous year increases the under-five mortality rate for the current year by 0.01% at a 1% significance level. This result aligns with Dhrifi (2020) [41], Langnel and Buracom (2020) [26], and Hadipour, Delavari, and Bayati (2023) [40], who argue that infant mortality is persistent in developing economies. This is attributable to limited access to healthcare services and infrastructure and shortages of healthcare facilities and trained personnel, especially in remote areas of SSA.

The results show that public health expenditure and external health expenditure have a negative impact on the U5MR. Thus, a 1% increase in public and external health expenditure is associated with a 0.1% and a 0.03% decrease in the under-five mortality rate at a 10% significance level. This indicates that allocating a reasonable proportion of public health expenditure enhances lower under-five mortality rates in SSA countries. The negative effect of external health on the U5MR is attributable to the fact that external health expenditure can provide additional funding for healthcare infrastructure, services, and programs, improving child survival rates. The results align with findings by [23,24,28,39,40,81].

Contrariwise, the domestic private health expenditure is positively signed. Thus, a percentage increase in HEDP worsens the U5MR by 0.2% at 5% significance levels. However, the out-of-pocket health expenditure has a positive but insignificant effect on the U5MR. Therefore, a positive impact of domestic private health expenditure on the U5MR suggests that personal spending typically serves as a coping mechanism when public healthcare systems are insufficient, compromising child healthcare and exacerbating child mortality. The finding aligns with Kulkarni (2016) [42] and Logarajan et al. (2022) [39]. Out-of-pocket expenditures harm infant mortality by potentially undermining health-seeking behaviors [44]. The positive effect of out-of-pocket health expenditure on the U5MR is attributable to high out-of-pocket healthcare expenses, which can act as a significant financial barrier, preventing many families from seeking timely medical care for their sick children. When parents or caregivers cannot afford necessary healthcare services, children may not receive prompt treatment, leading to worse health outcomes.

Empirical results show that a percentage increase in the prevalence of HIV and maternal mortality increases U5MR by 0.1% and 0.3–0.5%, respectively, at a 1% significance level. The positive effect of HIV on the U5MR accords with Anyanwu and Erhijakpor (2009) [22], Chihana et al. (2013) [69], Fowkes et al. (2016) [82], and Kiross et al. (2021) [27] who suggest that the prevalence of HIV reduces the chances of child survival. Studies such as Akinlo and Sulola (2019) [5] and Chewe and Hangoma (2020) [25] argue that HIV has a negative effect on under-five mortality due to an increase in the provision of antiretroviral during pregnancy, which has reduced the risk of mother-to-child transmission. However, the positive impact on the current empirical research is because of the lack of awareness and high HIV stigma in society and communities. Thus, without proper HIV awareness and growing HIV stigma amongst women, especially young people, young mothers become ignorant and afraid to expose their HIV status while pregnant, which leads to them infecting the unborn child, which may cause infant health complications, resulting in increased under-five mortalities. Again, limited access to medication exacerbates the under-five mortality rates in the region.

Regarding maternal mortality, the child is at risk of losing a primary caregiver when its mother dies. This can jeopardize the child’s access to necessary care, nutrition, and emotional support, all of which are essential for a child’s general development and survival. The positive effect of maternal mortality is consistent with Finlay et al. (2015) [75], Moucheraud et al. (2015) [76], and Scott et al. (2017) [83], who opine that maternal mortality compromises child survival. The results show that the prevalence of undernourishment is positively signed. Thus, a 1% change in undernourishment prevalence is associated with a 0.1% to 0.2% increase in the U5MR. This implies that the prevalence of undernourishment is an under-five mortality-promoting factor in sub-Saharan countries. The result aligns with Ssozi and Amlani (2015) [44] and Djoumessi (2022) [84], who suggest that undernourishment aggravates the mortality rate.

The results show that immunization has an asymmetric effect on under-five mortality in sub-Saharan countries. A 1% increase in immunization reduces the U5MR by 0.2% at a 10% significance level. The negative impact of immunization accords with Arthur and Oaikhenan (2017) [43], Akinlo and Sulola (2019) [5], Dhrifi (2020) [41], and Ayipe and Tanko (2023) [28], who posit that immunizations and other public health services shield children from harmful illnesses like polio, diphtheria, measles, and tetanus. Thus, immunization of children against these diseases lowers their risk of contracting and developing infections, which reduces the under-five mortality rate in SSA. A positive effect of immunization on under-five mortality could be attributed to low immunization and inequities in vaccination coverage [85] rates within the SSA population.

Again, a percentage increase in education reduces the U5MR by 0.1–2% at a 1% significance level. This implies that improved education for women makes women more knowledgeable about maternal health, including safe delivery techniques, appropriate nutrition throughout pregnancy, and the significance of prenatal and postnatal care. The result aligns with Grossman’s (1972, 2000) [29,30] theory, which postulated that education positively affects health demand. A negative impact of education on under-five mortality aligns with Ouedraogo et al (2020) [45], Owusu et al. (2021) [38], Ouedraogo, Simon and Kiragu (2022) [86] and Moradhvaj and Samir (2023) [87], who documented evidence of the negative impact of education on infant mortality. Once more, postponing parenthood can lower the chance of risks during pregnancy and delivery, which can help to improve results for the health of both mother and child.

The result indicates that ICT is an U5MR-reducing factor in SSA countries. Thus, a 1% increase in ICT reduces the U5MR by 0.12–0.4% at a 5% significance level. This implies that increased ICT can support telemedicine services, enabling medical professionals to diagnose and treat patients, including children, from a distance. A health information system that tracks and monitors child health indicators assists in identifying health trends and better allocating resources, ensuring that healthcare interventions are provided to the children most in need. Additionally, ICT can offer pregnant mothers access to training, as well as educational materials to nurses and community health workers, improving health outcomes and reducing under-five mortality rates. This result accords with Dutta et al. (2019) [50], Kouton, Bétila, and Lawin (2021) [51], and Khelfaoui et al. (2022) [88] who postulate that ICT is an infant mortality-reducing factor.

The results also show that a percentage increase in income per capita reduces the U5MR by 0.1–0.4% at a 5% significance level. This entails that a growing GDP per capita can help reduce poverty levels and favor better nutrition and adequate housing, which ultimately improves health capital. Therefore, families have more resources to invest in their children’s health, education, living conditions, and well-being [47]. The result corroborates with Makuta and O’Hare (2015) [23], Boachie and Ramu (2016) [89], Akinlo and Sulola (2019) [5], Dhrifi (2020) [41], and Logarajan et al. (2022) [39], who postulate that GDP per capita is associated with low under-five mortality rates. The mediating impact of institutional quality on the U5MR-health expenditure nexus is presented in the following section.

Table 3 shows that the institutional quality variable is positively signed. Thus, a 1% increase in institutional quality is associated with a 0.03–0.1% increase in the U5MR at a 10% significance level. This result is inconsistent with the expected a priori of the study. The positive effect could be attributable to weak institutional quality, which challenges the healthcare service delivery [40] and contributes to high child mortality. The following section shows the results of the impact of institutional quality on the U5MR-health expenditure nexus in SSA countries in Table 4.

#### The Role of Institutional Quality

Table 4 presents the empirical results of the impact of institutional quality on the under-five mortality rate and health expenditure relationship. The main focus of the results is on interaction terms in the last eight rows. Thus, Model 1 is the benchmark model that presents the institutional quality’s effect on the *U5MR*-public health expenditure as a percentage of GDP (*PHE*) nexus. Models 2 to 4 show results of the impact of institutional quality on the *U5MR*-health expenditure as measured by domestic private health expenditure (*HEDP*), out-of-pocket health expenditure (*HEOUP*), and external health expenditure (*HEEXT*), respectively.

**Table 4 ijerph-21-00333-t004:** Results of the Role of Institutional Quality on the U5MR-Health Expenditure Nexus.

	(Model_1)	(Model-2)	(Model_3)	(Model_4)
	(SGMM)	(SGMM)	(SGMM)	(SGMM)
Variables	lU5MR	lU5MR	lU5MR	lU5MR
L.U5MR	0.00878 ***	0.00404 *	0.00860 ***	0.00621 ***
	(0.00180)	(0.00234)	(0.00113)	(0.00195)
dlphiv	0.179	0.241	1.035 ***	−0.264
	(0.596)	(0.401)	(0.343)	(0.308)
dlpun	0.116	0.0432	0.427 ***	0.117
	(0.0827)	(0.0749)	(0.139)	(0.0835)
limm	0.0588	−0.0800	0.230	−0.159 *
	(0.153)	(0.145)	(0.232)	(0.0919)
lmam	0.425 ***	0.589 ***	0.404 ***	0.495 ***
	(0.0871)	(0.0879)	(0.0959)	(0.0866)
lsenr	0.207 *	−0.0283	−0.0654 *	−0.153
	(0.109)	(0.0878)	(0.0323)	(0.147)
dlict	−0.00629	−0.0225	−0.206 **	−0.0252
	(0.0272)	(0.0281)	(0.0863)	(0.0265)
dlgdppc	−0.134 *	−0.0915	−0.123	−0.118 **
	(0.0744)	(0.250)	(0.0808)	(0.0513)
linst	0.344 ***	−0.378 **	0.0237	0.170 **
	(0.0922)	(0.1636)	(0.0470)	(0.0756)
**Interaction Terms**				
lphe	−0.00921			
	(0.0632)			
lphelinst	−0.127 ***			
	(0.0432)			
lhedp		−0.0442		
		(0.0616)		
lhedplinst		0.103 *		
		(0.0551)		
dlheoup			−0.0854 *	
			(0.0490)	
dlheouplinst			−0.0164	
			(0.0274)	
lheext				−0.0416 *
				(0.0217)
lheextlinst				−0.0306 *
				(0.0180)
Constant	1.273 *	1.488 ***	3.034 *	1.291 ***
	(0.5234)	(0.3980)	(1.537)	(0.3490)
Observations	327	327	327	327
Number of countries	46	46	46	46
Number of instruments	26	27	28	26
F-Statistics *p*-value	0.000	0.000	0.000	0.000
AR(1) *p*-value	0.183	0.218	0.040	0.469
AR(2) *p*-value	0.102	0.515	0.100	0.141
Sargan test *p*-value	0.100	0.864	0.110	0.978
Hansen test *p*-value	0.873	0.706	0.567	0.439

Standard errors in parentheses. *** *p* < 0.01, ** *p* < 0.05, * *p* < 0.1.

Model 1 in Table 4 indicates the coefficient of the interaction term of public health expenditure (PHE) with institutional quality and external health expenditure (HEEXT) with institutional quality are negatively signified. The negative coefficient of the interaction terms indicates that institutions increase the negative effect of public and external health expenditure on the U5MR. This result is consistent with Makuta and O’Hare (2015) [23], Dhrifi (2020) [41], Langnel and Buracom (2020) [26], Dianda and Ouedraogo (2021) [47], and Hadipour, Delavari and Bayati (2023) [40], who indicate that improving institutional quality contributes to the development of democratic and meritocratic systems and efficient taxes which improves administrative capacity, thereby enhancing access to public and external health care and reducing infant mortality. This implies that public and external health expenditures are more likely to be effectively utilized in well-functioning healthcare systems with solid institutions, improving child health outcomes in SSA countries. Model 2 in Table 4 shows that the direct effect of institutional quality becomes negative after considering the interaction variable of domestic private health expenditure and institutional quality. Thus, a 1% increase in institutional quality reduces the U5MR by 0.4% at a 5% significance level. This implies that strong institutional quality can encourage greater healthcare utilization. This result aligns with Ouedraogo, Dianda and Adeyele (2020) [45], who suggest that institutional quality is relevant in improving health outcomes in the SSA region. Model 3 shows a marginal negative effect of out-of-pocket health expenditure on the U5MR.This implies that an increase in out-of-pocket health expenditure is associated with a decrease in U5MR, ceteris paribus. This aligns with Kimani (2014) [90] who suggests that out-of-pocket health expenditure helps households to restore health.

The interaction term of HEDP with institutional quality is positively signified. This implies that institutional quality enhances the positive impact of domestic private expenditures on the U5MR rate. The positively signed interaction term coefficient could be due to poor and weak institutional quality [35,41], which may lead to chronic underinvestment in public healthcare systems and a lack of healthcare infrastructure. This deficiency forces individuals and families to turn to private healthcare providers, incurring substantial out-of-pocket expenses for what they perceive as better-quality care [90,91]. This can leave marginalized communities with no choice but to rely on private providers, even if it leads to a financial burden, worsening health outcomes, and higher mortality rates.

## 5. Summary, Conclusions, and Policy Recommendations

The study investigated the impact of health expenditure on under-five mortality rates (U5MRs) in 46 sub-Saharan African (SSA) countries and considered the role of institutional quality by employing a panel data estimation technique using the System GMM estimator. The study analyzed four categories of health expenditure: public health expenditure, domestic private health expenditure, out-of-pocket health expenditure, and external health expenditure. The study also examined various socio-economic and health-related factors. The empirical results reveal several key findings.

The research emphasized the pivotal role of public health expenditure and external funding in reducing child mortality. Simultaneously, it uncovered the adverse impact of domestic private and out-of-pocket health expenditure on the U5MR. Notably, the prevalence of HIV and maternal mortality emerged as significant factors detrimental to child health. Moreover, the study underscores the potential of child immunization, education, and Information and Communication Technology (ICT) to mitigate the U5MR effectively. Economic development is found to positively influence child health. However, what stands out prominently is the central role of institutional quality. Strong institutions enhance the efficiency of health expenditure, translating into a reduction in child mortality, whereas weaker institutional quality often steers individuals toward private healthcare, exacerbating child mortality.

In light of these findings, the foremost recommendation for SSA countries is to channel their efforts into two primary areas: enhancing public health expenditure and strengthening institutional quality. Governments should allocate a larger share of their budgets to bolster public health services, particularly in marginalized and underserved regions. Concurrently, a dedicated focus should be on reforming and reinforcing institutions that oversee healthcare delivery. These reforms should emphasize transparency, accountability, and efficiency in allocating healthcare resources, effectively curbing corruption.

Furthermore, initiatives to curb the prevalence of HIV and maternal mortality should be intensified. This necessitates comprehensive awareness campaigns and increased accessibility to critical medical services, particularly for expectant mothers. Simultaneously, it is imperative to promote child immunization and education, especially for women, as an integral approach to enhancing child and maternal health.

The integration of ICT into healthcare systems should be prioritized, as it can significantly enhance the accessibility and quality of healthcare services, facilitating effective monitoring and timely interventions. Lastly, governments should focus on economic development, as higher income per capita is intrinsically associated with reduced U5MRs.

Governments and international organizations should prioritize institutional strengthening to enhance healthcare in SSA, ensuring transparency and efficient resource allocation. This entails capacity building for healthcare professionals, robust regulatory frameworks to combat corruption, collaboration with international partners for knowledge sharing, and ongoing evaluation to drive improvements in healthcare governance.

Future research should investigate the mechanisms through which institutional quality affects health expenditure and child mortality, assess the effectiveness of policy interventions, and explore the impact of cultural and sociodemographic factors on child health within the context of healthcare systems in SSA. These research directions will yield valuable evidence-based policy recommendations for enhancing child health, emphasizing the role of institutions.

## 6. Limitations and Recommendations for Future Research

This study is limited by factors such as the availability and quality of data, which may have restricted the inclusion of certain variables and affected the accuracy of the analysis. To address these limitations and deepen our understanding of under-five mortality, future research should consider longitudinal studies to track trends over time. Exploring the impact of sociodemographic factors, such as adolescent pregnancies and health awareness, through qualitative research methods could provide valuable insight. Comparative analyses across different countries or regions with varying levels of institutional quality and healthcare systems can identify best practices and inform evidence-based policy recommendations. Additionally, intervention studies evaluating the effectiveness of targeted interventions to improve institutional quality and healthcare access among children under five are essential for informing policy decisions and effectively reducing under-five mortality rates.

## Figures and Tables

**Table 1 ijerph-21-00333-t001:** Variable Definitions, Expected A priori, and Sources.

Variable	Definition	Expected A priori	Source
U5MR	Under-five mortality rate: the probability per 1000 that a newborn baby will die before age five if subject to age-specific mortality rates of the specified year.		Global Economy Database (GED)
phe	Public health expenditure: level of current health expenditure expressed as a percentage of GDP. Estimates of current health expenditures include healthcare goods and services consumed each year.	Negative	GED
hedp	Domestic private health expenditure: share of current health expenditures funded from domestic private sources, including funds from households, corporations, and non-profit organizations. The spending can be either prepaid to voluntary health insurance or paid directly to healthcare providers.	Negative	World Bank Database
heoup	Out-of-pocket health expenditure: share of out-of-pocket payments of total current health expenditures. Out-of-pocket payments are spending on health directly out-of-pocket by households.	Negative	World Bank Database
heext	External health expenditure: current external expenditures on health per capita expressed in international dollars at purchasing power parity. External sources are composed of direct foreign transfers and foreign transfers distributed by the government, encompassing all financial inflows into the national health system from outside the country.	Negative	World Bank Database
phiv	Prevalence of HIV: the percentage of people aged 15–49 who are infected with HIV.	Positive	GED
pun	Prevalence of undernourishment: the percentage of the population whose habitual food consumption is insufficient to provide the dietary energy levels that are required to maintain a normal, active, and healthy life.	Positive	GED
imm	Child immunization against DPT: the percentage of children ages 12–23 months who received DPT vaccinations before reaching 12 months.	Negative	GED
mam	Maternal mortality ratio: the number of women who die from pregnancy-related causes while pregnant or within 42 days of pregnancy termination per 100,000 live births.	Positive	GED
senr	Gross enrollment ratio: the ratio of total enrollment to the age group population that officially corresponds to the level of education shown. Secondary education completes the provision of basic education that begins at the primary level and aims to lay the foundations for lifelong learning and human development by offering more subject- or skill-oriented instruction using more specialized teachers.	Negative	GED
ict	Technology index value: generated through PCA and computed using the PCA from fixed broadband subscription, internet use, and mobile phone usage data.	Negative	Author’s calculations of ICT composites with data from the World Bank Database using PCA on Stata
gdppc	GDP per capita: GDP divided by midyear population. GDP is the sum of gross value added by all resident producers in the economy plus any product taxes and minus any subsidies not included in the value of the products.	Negative	World Bank Database
inst	Institutional index: computed using PCA from six indicators of institutional quality, including rule of law, government effectiveness, control of corruption, regulatory quality, voice accountability, and political stability data—the value of each of the six components ranges from −2.5 weak to 2.5 strong.	Negative	Author’s calculations of institutional quality composites with data from the GED database using PCA on Stata

**Table 3 ijerph-21-00333-t003:** Direct Impact of Health Expenditure on the U5MR.

	(Model_1)	(Model-2)	(Model_3)	(Model_4)
	(SGMM)	(SGMM)	(SGMM)	(SGMM)
Variables	lU5MR	lU5MR	lU5MR	lU5MR
L.U5MR	0.00798 ***	0.00896 ***	0.00641 ***	0.00615 ***
	(0.00149)	(0.00233)	(0.00143)	(0.00159)
dlphiv	0.129 ***	0.126 ***	−0.341	0.0721 ***
	(0.0252)	(0.0375)	(0.439)	(0.0213)
d.lpun	0.229 *	0.0742	0.189 *	0.0996 *
	(0.150)	(0.0940)	(0.0930)	(0.0505)
Limm	0.379 *	−0.205	−0.178 *	0.180 **
	(0.230)	(0.150)	(0.0913)	(0.0694)
Lmam	0.415 ***	0.314 ***	0.452 ***	0.538 ***
	(0.101)	(0.0934)	(0.0695)	(0.0839)
Lsenr	−0.0914 *	−0.0578	0.0211	−0.213 ***
	(0.0466)	(0.122)	(0.0678)	(0.0640)
dlict	−0.352 **	−0.0475	−0.0175	−0.117 ***
	(0.134)	(0.0287)	(0.0173)	(0.0342)
dlgdppc	−0.0552 **	−0.176 **	−0.368 **	−0.293
	(0.0239)	(0.0719)	(0.178)	(0.247)
Linst	0.00730	0.0566 *	0.0336 *	0.0429 *
	(0.0456)	(0.0288)	(0.0169)	(0.0247)
Lphe	−0.131 *			
	(0.0657)			
Lhedp		0.152 **		
		(0.0643)		
dlheoup			0.0208	
			(0.153)	
Lheext				−0.0341 *
				(0.0167)
Constant	0.418 ***	3.007 ***	1.097	−0.0447
	(0.1336)	(1.001)	(0.750)	(0.669)
Observations	327	327	327	327
Number of countries	46	46	46	46
Number of instruments	26	25	27	28
F-Statistics p-value	0.000	0.000	0.000	0.000
AR(1) p-value	0.127	0.100	0.410	0.264
AR(2) p-value	0.100	0.123	0.132	0.416
Sargan test p-value	0.666	0.770	0.687	0.667
Hansen test p-value	0.873	0.684	0.445	0.529

Standard errors in parentheses. *** *p* < 0.01, ** *p* < 0.05, * *p* < 0.1.

## Data Availability

Publicly available datasets were analyzed in this study. This data can be found here [https://www.theglobaleconomy.com/ and https://databank.worldbank.org/source/world-development-indicators].

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
