# Peer review of "Health Expenditure, Institutional Quality, and Under-Five Mortality in Sub-Saharan African Countries"

_ijerph, 2024, doi:10.3390/ijerph21030333_

Round 1

Reviewer 1 Report

Comments and Suggestions for Authors

Through this manuscript the authors explored the complex relationship between health expenditure and under-five mortality rates, focusing on the mediating role of institutional quality by employing a systems GMM model. The analysis is based on data collected between 2000 and 2021 in 46 sub-Saharan African countries.

The manuscript is well conceived, and executed satisfactorily. However, there are certain comments /suggestions which need to be considered to make it in acceptable form.

1.       The first sentence in the abstract should be changed. The study was not conducted between 2000 and 2021. The study used data between these periods.

2.       In introduction, Lines 46-48 mentions “In Sub-Saharan Africa, where healthcare resources are often scarce, and disparities are pronounced, the relationship between health expenditure and under-five mortality rates has emerged as a critical area of concern -  What about the impact of health spending on infant mortality and neonatal mortality? What are their relative share? Why Under5 mortality alone is addressed? Please justify?

3.       The concept “institutional quality” (as per theoretical literature page 2 and 3)  is used as mediating factor. However, the definition and measurement of variables such as rule of law, government effectiveness, control of corruption, regulatory quality, voice accountability, and political stability data considered for institutional quality is unclear. – Please provide details of institutional quality variables used in the study.

4.       Line 208 mentions —the values of the six components range from -2.5 weak to 2.5 strong. – How was this calculated. Here more details required.

5.       There are many other sociodemographic factors such as adolescent pregnancies, health awareness, etc., inextricably impacting under five mortality, which are missed in discussion.  

6.       Allocation of government resources on curative vs. primary care also impact the under 5 mortalities. Few countries allocate more budget on primary care compared to others. In other words, allocative efficiency of existing resources in countries would also influence the health outcomes like under-five mortality in countries – may include in discussion/limitations.

7.       In few places, sources are required. For example in lines 113-114 - Different conclusions were generated as other studies document evidence of no association between health spending (???).

8.       A brief paragraph on limitation should be included before conclusion.

Minor corrections

a)       Line 288-289 - The sentence should be corrected. It should be 90.2 per 1000 births.

b)      Line 296-97 – Please correct the year in the sentence (between 2000 and 2021)

c)       Line 324-325 –Please correct the repetition in the sentence

d)      Line 214 – correct the sentence

e)      The manuscript need English editing  

Comments on the Quality of English Language

  English Language editing required

Author Response

Dear Prof

Thank you for reviewing our manuscript. All the changes made to the revised document are highlighted in yellow.

Kind Regards

Dorcas

Reviewer 2 Report

Comments and Suggestions for Authors

Author Response

(The authors gave the same response as above.)

Round 2

Reviewer 2 Report

Comments and Suggestions for Authors

The paper has been improved . It can be accepted in present form for publication.